# Precipitable Water Vapor Retrieval from Shipborne GNSS Observations on the Korean Research Vessel ISABU

**DOI:** 10.3390/s20154261

**Published:** 2020-07-30

**Authors:** Dong-Hyo Sohn, Byung-Kyu Choi, Yosup Park, Yoon Chil Kim, Bonhwa Ku

**Affiliations:** 1Space Science Division, Korea Astronomy and Space Science Institute, Daejeon 34055, Korea; dhsohn5@gmail.com; 2Korea Institute of Ocean Science and Technology, Busan 49111, Korea; yosup@kiost.ac.kr (Y.P.); yckim@kiost.ac.kr (Y.C.K.); bhku@kiost.ac.kr (B.K.)

**Keywords:** GNSS, shipborne, precipitable water vapor, radiosonde, AIRS, kinematic precise point positioning

## Abstract

We estimate precipitable water vapor (PWV) from data collected by the low-cost Global Navigation Satellite System (GNSS) receiver at a vessel. The dual-frequency GNSS receiver that the vessel ISABU is equipped with that is operated by the Korea Institute of Ocean Science and Technology. The ISABU served in the Pacific Ocean for scientific research during a period from August 30 to September 21, 2018. It also performs radiosonde observations to obtain a vertical profile of troposphere on the vessel’s path. The GNSS-derived PWV is compared to radiosonde observations and the Atmospheric Infrared Sounder (AIRS) on NASA’s Aqua satellite output. A bias and root-mean-square (RMS) error between shipborne GNSS-PWV and radiosonde-PWV were −1.48 and 5.22 mm, respectively. When compared to the ground GNSS-PWV, shipborne GNSS-PWV has a relatively large RMS error in comparison with radiosonde-PWV. However, the GNSS observations on the vessel are still in good agreement with radiosonde observations. On the other hand, the GNSS-PWV is not well linearly correlated with AIRS-PWV. The RMS error between the two observations was approximately 8.97 mm. In addition, we showed that the vessel on the sea surface has significantly larger carrier phase multipath error compared to the ground-based GNSS observations. This also can result in reducing the accuracy of shipborne GNSS-PWV. However, we suggest that the shipborne GNSS has sufficient potential to derive PWV with the kinematic precise point positioning (PPP) solution on the vessel.

## 1. Introduction

Precipitable water vapor (PWV) in the troposphere is an essential parameter for massive rain prediction, climate changes, and Global Navigation Satellite System (GNSS) positioning. For the PWV in the troposphere, there are many types of observing systems, such as GNSS, radiosonde, microwave radiometer, space-based satellites with thermal infrared and passive microwave imagery. The GNSS is a useful monitoring tool to provide accurate water vapor for 24 h regardless of the weather conditions. It also has some advantages, such as high temporal resolution and long-term stability. The GNSS-PWV has been primarily carried out using the ground-based observations. The ground-based GNSS is capable of producing continuous and highly accurate PWV. To verify the accuracy of the ground GNSS-PWV, some studies compared PWV obtained from different observations, such as radiosonde [1,2,3], microwave radiometer [4,5,6], satellite measurements [7,8,9], Raman Lidar measurements [10,11]. The accuracy of the ground-based GNSS-PWV is approximately 1–5 mm in root mean square (RMS). Similar results on the Korean Peninsula have also been reported in several studies [12,13,14].

GNSS-PWV retrieval was limited to the use of ground-based reference stations. Recently, however, some studies reported that the shipborne GNSS equipment over the ocean is useful for monitoring the PWV. The PWV estimated by shipborne GNSS in the open ocean is in good agreement with radiosonde observations [15,16,17]. Boniface et al. [18] performed the comparison between PWV derived from shipborne GNSS, the output from the numerical weather prediction model, and the Moderate Resolution Imaging Spectroradiometer (MODIS) retrieval over the Mediterranean Sea. They reported that there are no significant PWV biases between GNSS, MODIS, and a model. However, statistics indicated that the root-mean-square (RMS) differences between the two observations are a magnitude of 2.6 mm and 3.4 mm, respectively. They also suggested that the shipborne GNSS is beneficial for observing accurate PWV. Fan et al. [19] estimated the PWV from GNSS equipment mounted on a lightweight ship in the Chinese Bohai Sea. They showed that GNSS-derived PWV is well coincident with the fifth-generation Penn State/NCAR Mesoscale Model. In addition, Shoji et al. [20,21] compared the results of the shipborne GNSS-derived PWV to radiosonde observations in the western North Pacific and the seas adjacent to Japan. In 2016, they reported that the shipborne GNSS-PWVs have RMS errors of 3.4–5.4 mm compared to radiosonde observations. In 2017, the difference in GNSS retrieved-PWVs versus radiosonde observations was compared to several components, such as the atmospheric delay, the altitude of the GNSS antenna, the vessel’s speed, the wind speed, and the wave height. They suggested that GNSS is useful in monitoring PWV in oceans around the world. Recently, Wang et al. [22] investigated that the PWV derived from shipborne multi-GNSS, including GPS, GLONASS, and Galileo, is in good agreement with a numerical weather model and radiosonde observations. They also showed that the shipborne multi-GNSS-PWV is consistent with a satellite altimetry PWV in the Arctic Ocean.

This study investigates the GNSS-PWV from the shipborne GNSS observations for 23 days in in the Pacific Ocean. We focus on calculating PWV using the data obtained from the low-cost GNSS receiver on the Korean research vessel. The GNSS kinematic PPP method is adopted for PWV retrievals with a moving vessel. To validate retrieved PWV from shipborne GNSS observations, we compare the GNSS-derived PWV to radiosonde observations and Atmospheric Infrared Sounder (AIRS) aboard NASA’s Earth-observing system satellite output. Statistical results are presented between the two different observations. In addition, a linear regression method is considered to measure the correlation between pairs of variables (GNSS and radiosonde, GNSS and AIRS).

## 2. Data and Processing

### 2.1. Shipborne GNSS Data

To calculate the PWV from shipborne GNSS observations over the ocean, GNSS data obtained from the ocean science research vessel ISABU of the Korea Institute of Ocean Science and Technology (KIOST) are considered. The ISABU is Korea’s marine science research vessel with a 5894-ton and 99.8 m in length [23]. It is equipped with a motion reference unit (Kongsberg Maritime MRU5) that can get 6 axis motion with 20 Hz. This vessel is also equipped with the low-cost GNSS equipment (C-Nav3050 receiver and C-Nav286 antenna) and a radiosonde observation facility. It can also provide continuous GNSS data with sampling intervals of 30 s. The vessel ISABU operated in the Pacific Ocean for scientific research during a period from day-of-year (DOY) 242 (30 August 2018) to DOY 264 (21 September 2018). Weather conditions such as wind, temperature, relative humidity, and air pressure were generally stable for this period. However, a severe weather condition associated with strong wind and low pressure was observed for two days (5 September and 11 September 2018). An averaged rolling variation of the ISABU was about ± 2 degrees for the same period. The red circle in Figure 1 represents the location of the GNSS antenna installed on the mast of the vessel. The height of the GNSS antenna from the sea surface is approximately 35.14 m.

GNSS kinematic precise point positioning (PPP) method is adopted to estimate the ISABU’s moving paths and the tropospheric zenith wet delays (ZWD). In general, GNSS PPP uses dual-frequency measurements to obtain a more precise position solution. In this study, we used a multi-GNSS analysis software (MGAS) developed by the Korea Astronomy and Space Science Institute [24]. MGAS is possible to process GNSS data for PPP. In addition, it can support some satellite constellations, such as GPS, GLONASS, Galileo, BeiDou, and QZSS [25]. To remove the ionospheric delay errors, the ionospheric-free code and phase linear combinations can be expressed by the following Equations (1) and (2).
(1)$$P_{IF}=\rho +c\;\cdot \;dt-c\;\cdot \;dT+d_{trop}+\varepsilon_{P_{IF}}$$
(2)$$\Phi_{IF}=\rho +c\;\cdot \;dt-c\;\cdot \;dT+d_{trop}+\lambda N_{IF}+\varepsilon_{\Phi_{IF}}$$
where $P_{IF}$ and $\Phi_{IF}$ are the ionospheric-free code observables and ionospheric-free phase observables, respectively. ρ is the geometric range from the GNSS satellite to the ground receiver, c is the speed of light, and dt and dT are the receiver and the satellite clock errors, respectively. $d_{trop}$ is the tropospheric delay error, λ is the ionospheric-free wavelength, $N_{IF}$ is the ionospheric-free phase ambiguity. $\varepsilon_{P_{IF}}$ and $\varepsilon_{\Phi_{IF}}$ represent the code and phase measurement noises, respectively.

An extended Kalman filter (EKF) is applied for PPP parameter estimation. The update and prediction steps of the state variables for PPP are expressed as Equations (3) and (4).
(3)$$\{\begin{array}{c} K_{k}=P_{k}\left(-\right)H_{k}^{T}{\left(H_{k}P_{k}\left(-\right)H_{k}^{T}+R_{k}\right)}^{-1}\\ {\hat{X}}_{k}\left(+\right)={\hat{X}}_{k}\left(-\right)K_{k}\left(z_{k}-h\left({\hat{X}}_{k}\left(-\right)\right)\right)\;\\ P_{k}\left(+\right)=\left(I-K_{k}H_{k}\right)P_{k}\left(-\right)\; \end{array}$$
(4)$$\{\begin{array}{c} {\hat{X}}_{k+1}\left(-\right)={\hat{X}}_{k}\left(+\right)+\int_{t_{k}}^{t_{k+1}}f\left({\hat{X}}_{k}\left(+\right),\;\tau \right)d\tau \;\;\\ P_{k+1}\left(-\right)=\Phi \left(t_{k+1},t_{k}\right)P_{k}\left(+\right)\Phi {\left(t_{k+1},t_{k}\right)}^{T}+Q_{k} \end{array}$$
where ${\hat{X}}_{k}$ and $P_{k}$ represent the state variable vector and the variance-covariance matrix, respectively. $K_{k}$ denotes the Kalman gain. $H_{k}$ is the design matrix, $z_{k}$ is the actual measurement. $R_{k}$ and $Q_{k}$ are the process noise and measurement noise, respectively. Φ denotes the state transition matrix. The unknown state vector in PPP processing can be written as follows: (5)$$\vec{X}=\left(x,\;y,\;z,\;dt,\;ZWD,\;trop_{G_{N}},\;trop_{G_{E}},\;N_{1\ldots m}\right)$$
where the unknown state vector, $\vec{X}$, includes the receiver coordinates (x,y,z) the receiver clock error (dt), ZWD, the horizontal tropospheric gradients $\left(trop_{G_{N}}\;and\;trop_{G_{E}}\right)$, and the ambiguities $\left(N_{1\ldots m}\right)$ as the float solution. To achieve precise positioning results, corrections such as tide effects, phase wind up, antenna phase center offsets (PCO)/phase center variations (PCV), and Sagnac effect have to be considered using models. A more detailed explanation for the GNSS kinematic PPP models is listed in Table 1.

The tropospheric delay can be divided into the hydrostatic and wet delay components. In addition, the horizontal gradient components can be considered [26].
(6)$$d_{trop}=m_{h}\;\cdot \;ZHD+m_{w}\;\cdot \;ZWD+m_{G}\;\cdot \;\left\{G_{N}\;\cdot \;\cos\left(Az\right)+G_{E}\;\cdot \;\sin\left(Az\right)\right\}$$
where ZHD and ZWD represent the zenith hydrostatic and wet delays, respectively. $m_{h}$ and $m_{w}$ denote the hydrostatic and wet mapping functions. In this paper, the global mapping function (GMF) based on the European Centre for Medium-Range Weather Forecasts numerical model is applied to convert the tropospheric delay errors from zenith direction to the slant direction. $m_{G}$ denotes the gradient mapping function, $G_{N}$ and $G_{E}$ are the horizontal gradients in the north and east direction, respectively. Az is the azimuth angle of a GNSS satellite.

For the PPP tropospheric error estimation, a prior ZHD is calculated by the Saastamoinen model that is based on meteorological information derived from the Global Pressure and Temperature model 2 [27]. Usually, *ZHD* with the Saastamoinen model can be expressed as Equation (7).
(7)$$ZHD=\frac{0.0022768\;\cdot \;P_{s}}{1-0.00266\cos\left(2\varphi \right)-0.00028\;\cdot \;H}$$
where $P_{s}$ is the total surface pressure, φ and H represent the coordinate (latitude and ellipsoidal height) of the station, respectively.

The ZWD, $G_{N}$, and $G_{E}$ are estimated as the unknown parameters with the other parameters in PPP. The ZWD is considered as a measure of the total PWV in the atmosphere.
(8)PWV= π ×ZWD
(9)$$\pi =\frac{10^{6}}{\rho_{w}\;\cdot \;R_{v}\left\{\left(\frac{k_{3}}{T_{m}}\right)+k_{2}\right\}}$$
where π is the conversion factor, which is a dimensionless quantity. $\rho_{w}$ represents the density constant of liquid water. $R_{v}$ is the gas constant for water vapor. The values of the refractivity constants are $k_{3}=\left(3.739\pm 0.0012\right)\;\cdot \;10^{5}\left(\frac{K^{2}}{hPa}\right)$, and $k_{2}=22.1\pm 2.2\;K/hPa$, respectively. $T_{m}$ is the mean weighted temperature of the atmosphere, which can be calculated by the formulation of Bevis et al. [28].

### 2.2. Radiosonde and Satellite Data

The radiosonde is a representative system for measuring in situ various atmospheric parameters (temperature, humidity, air pressure, etc.) in the atmosphere. A radiosonde is a battery-powered telemetry instrument carried into the atmosphere, usually by a balloon. It transmits measured data which is received by the antenna to a ground receiver by radio. A GPS antenna attached to the instrument provides the system with positional information that is used for applying differential corrections to the wind data and also for determining the position of a moving sounding station. The total precipitable water, WV, is that contained in a column of unit cross-section extending all of the ways from the Earth’s surface to the top of the atmosphere. The WV contained in a layer bounded by pressures p1 and p2 is given by Equation (10).
(10)$$WV=\frac{1}{g}\int_{p1}^{p2}MR\;dp$$
where g is the acceleration of gravity, MR represents a mixing ratio at the pressure level [30]. To calculate the mixing ratio, a saturated water vapor pressure needs to be computed for air temperature. In addition, it is necessary to calculate an actual water vapor pressure for dew point temperature. Radiosonde measurements include air temperature, a dew point temperature, and atmospheric pressure.

The AIRS aboard NASA’s Earth-observing system satellite, Aqua, is capable of measuring the water vapor over the oceans [31]. In this study, we use standard level 2 products (i.e., AIRS2RET). They can be downloaded from the website (https://www.image.ucar.edu/DAReS/DART/Manhattan/observations/obs_converters/AIRS/AIRS.html). The AIRS level 2 retrievals contain geophysical quantities, such as estimates of atmosphere and surface properties, profiles of temperature, water vapor, and ozone, total precipitable water vapor (PWV), etc. In geophysical quantities, the AIRS satellite’s PWV is the total column water vapor (unit: mm) from the surface to the upper troposphere. The uncertainty of the AIRS-PWV is within 5% in the intermediate range between 20 and 40 mm. On the other hand, it has a wet bias of more than 5% for low PWV values and a dry bias of more than 5% for high PWV values [32]. A 6-min AIRS granule does not exactly match the ship’s locations. Therefore, the averaged value of the PWV data is calculated within the latitude ± 0.5° and longitude ± 0.5 ° based on the location of the shipborne GNSS receiver from the AIRS dataset.

## 3. Comparison of PWVs on the Ground

To verify the PWV values derived from the GNSS kinematic PPP, the dual-frequency data obtained from the three GNSS stations on the ground was precisely processed. The GNSS equipment (receiver and antenna) used at the three stations are listed in Table 2. We compared GNSS-derived PWVs to radiosonde observations closest to each GNSS site. In addition, the retrieved PWVs from the AIRS satellite were also used to compare to GNSS PWVs.

Usually, radiosonde measure takes approximately 30 min to travel up through the significant paths with a broad distribution of water vapor in the troposphere [33]. In this study, GNSS-derived PWV was used as the averaged value for 30 min, starting with radiosonde observation. AIRS2RET dataset has a time resolution of 6 min [31]. For comparison of GNSS-PWV and AIRS retrieved-PWV, GNSS-PWV was adjusted to an averaged value for 6 min.

Figure 2 shows the three GNSS stations that are used to validate and compare the PWV estimated by the GNSS kinematic PPP. The green triangle represents the location of the GNSS stations; the red circle indicates the location of the radiosonde observation. The distances between the two different observations (GNSS and radiosonde) are all within 11 km. The radiosonde data can be available at the Department of Atmospheric Science at the University of Wyoming (http://weather.uwyo.edu/upperair/sounding.html).

Table 3 shows the statistical comparison of GNSS derived-PWV, radiosonde retrieved-PWV, and AIRS measurements. As shown in Table 3, the bias and RMS value of PWV were presented with increasing distance between the GNSS station and radiosonde observation. The Backryeong-do (BRDO) site with the shortest distance between the different observations has a bias of 1.27 mm, and an RMS value of 1.47 mm between GNSS derived-PWV and radiosonde-PWV. It is noted that the bias between the two observations at the Kwangju (KWNJ) site increased significantly. KWNJ has a distance of more than 10 km from the GNSS station to radiosonde observations. Therefore, the bias of the estimated PWV can be increased according to the distance between the two different observations. On the other hand, the RMS value in all stations was less than 5 mm. Similar to the bias pattern, RMS value also shows any significant change with increasing distance between the GNSS station and radiosonde observation. As the distance increases between two different observations, the RMS value also increases. Our results are similar to other ground-based results reported in earlier studies [1,2,3]. As the distance increases between the GNSS station and radiosonde observations, the bias of PWV tends to increase. It indicates that the distance between two different observations can significantly affect bias and RMS values.

We further compared the GNSS-PWV to the AIRS-PWV. As listed in Table 3, it can be seen that the difference between the GNSS-PWV and the AIRS-PWV is relatively large when compared to the results of the GNSS-PWV and radiosonde-PWV. Moreover, the AIRS-PWVs tend to be overestimated compared to the GNSS-PWV. He and Liu [9] investigated that the AIRS-PWV tends to be overestimated compared to radiosonde observations and GNSS retrieved-PWV. Therefore, our results agree well with the results reported by He and Liu [9]. The AIRS data do not exactly match the location of the GNSS station. Raja et al. [34] suggested that the AIRS retrieved-PWV depends on the observation locations and seasons. Furthermore, the significant bias of the AIRS-PWV can be closely related to the observation locations on the Korean Peninsula. It is noted that the PWV values derived from a kinematic PPP solution on the ground are reliable and precise.

## 4. Comparison of PWVs on the Ocean

### 4.1. Comparisons of GNSS and Radiosonde

The Korean ISABU vessel is equipped with a radiosonde observation instrument, the Vaisala RS41-SG. Radiosonde observations are usually done twice a day (0:00 and 12:00 UTC) on a moving vessel. In this study, we first calculated the PWV by processing the GNSS dual-frequency observations received from the ISABU vessel moving in the Pacific Ocean. The PWV derived from the GNSS kinematic PPP was compared to radiosonde retrieved-PWV in the ocean. Figure 3 shows the changes in PWVs derived from the GNSS observations along with the ISABU vessel’s trajectories. The GNSS-PWVs have a value in the range of 10 to 90 mm.

Figure 4 shows the results of comparing the GNSS derived-PWV and radiosonde retrieved-PWV during the period from DOY 242 to DOY 264, 2018. Some radiosonde observations launched from the vessel ISABU can be inaccurate due to the loss of signals from the radiosonde. It can have a significant impact on comparing PWVs from the two different observations. In this study, we have excluded radiosonde data with some signal losses. As shown in Figure 4, a linear regression model was considered to investigate the correlation between GNSS-PWV and radiosonde-PWV. The resulting line from the linear regression analysis was plotted on a scatter diagram. The absolute magnitude of the correlation coefficient (R) between the two variables was 0.85. According to a classification proposed by Schober et al. [35], this indicates that there is a high correlation (0.70–0.89) between the two variables in the Pacific Ocean.

Table 4 describes the PWV values derived from GNSS and radiosonde observations. The bias and RMS error between the two observations are −1.48 and 5.22 mm, respectively. A minimum and a maximum difference of GNSS-PWV and radiosonde-PWV are 0.17 and −10.81 mm. From these biases, it can be seen that GNSS-PWV is slightly underestimated compared to radiosonde-PWV. Although there are a few discrepancies between the two observations, the GNSS, however, is in good agreement with radiosonde observations. When compared to the ground-based GNSS-PWV, shipborne GNSS-PWV has a relatively large RMS error in comparison with radiosonde-PWV. Unlike the ground-based GNSS observations, the GNSS observations at the moving vessel are exposed to large error sources. In particular, there may be a problem with a GNSS antenna’s orientation due to the rolling of the vessel. Moreover, the vessel on the sea surface has a large multipath error that has been identified as a major source of error in many precise applications [36].

Figure 5 displays the time series of multipath errors at a ground station and the vessel ISABU. GPS L1 code multipath (MP1) error was calculated by TEQC software [37]. The MP1 errors on the sea surface and the ground are marked with blue dots and red dots, respectively. As shown in Figure 5, it can be seen that the L1 code MP1 error on the sea surface is significantly larger than that on the ground. RMS values of MP1 at sea and ground are 0.55 and 0.29 m, respectively. RMS values of MP1 error differ almost twice as much in different regions. Presumably, the sea surface can increase the probability of GNSS signal reflection [38,39]. Therefore, it also can result in reducing the accuracy of shipborne GNSS-PWV. Shoji et al. [21] reported that the GNSS antennas at different locations on the ship could affect PWV accuracy due to multipath errors. Our results clearly show the difference of MP1 between at the sea surface and at the ground.

Figure 6 shows the time series of GPS L3 carrier phase multipath errors on DOY 242, 2018. The carrier phase multipath errors on the sea surface and the ground are marked with a solid blue line and a solid brown line, respectively. As shown in Figure 6, the carrier phase multipath errors are very tiny compared to code MP1 errors. Similar to the code MP1 errors, it can be seen that the L3 carrier phase multipath error on the sea surface is much larger than that of the ground. In the BRDO station, the carrier phase multipath errors have a value ranging from −0.05 to 0.05 m. The RMS value of them is approximately 0.007 m. On the other hand, most of the errors in carrier phase multipath at the vessel ISABU changed from about −0.2 to 0.2 m. In addition, the carrier phase multipath errors reached up to about 1 m. The RMS value of them is approximately 0.068 m. The RMS values of carrier phase multipath errors between on the sea surface and the ground are about ten times different. As a result, the large carrier phase multipath errors at the vessel ISABU may have been a predominant error factor in calculating GNSS-PWV.

There have been several rapid movements of the vessel ISABU during radiosonde observations at the ship. Figure 7 shows the correlation between the absolute PWV difference (GNSS – Radiosonde) and vessel speed. The red dots indicate the absolute difference between GNSS-PWV and radiosonde-PWV for vessel speed. The solid black line represents the relationship between the two variables by fitting a linear regression approach. Where the vessel speed means the average velocity of the ship during radiosonde observations. As seen in Figure 7, although the correlation coefficient (R~0.42) shows a weak correlation between two variables, the relation tends to increase with the speed up. In addition, the RMS error between GNSS-PWV and Radiosonde-PWV was approximately 1.8 mm when the ship’s speed is below 6 knots. On the other hand, it was approximately 7.1 mm when the ship’s speed is above 6 knots. As a result, the rapid movement of the vessel can also make an enormous difference between GNSS-PWV and radiosonde-PWV.

### 4.2. Comparisons of GNSS and AIRS

The comparisons between GNSS and satellite observations are carried out for an experimental period in the Pacific region. In this study, we compared the PWV derived from the shipborne GNSS receiver to the PWV retrieved from the AIRS mounted on the Aqua satellite. Figure 8 shows a typical sample image of AIRS observations at the specific time and the location of the vessel ISABU at that time. AIRS-PWV was calculated by extracting observed values within the latitude ± 0.5° and longitude ± 0.5° from the position of the ship.

Figure 9 shows the comparison of GNSS-PWV and AIRS-PWV. The minimum and maximum differences between the two observations are 0.84 and 17.37 mm, as listed in Table 5, respectively. Its bias and RMS error are also about 0.21 and 8.95 mm, respectively.

As seen in Figure 9, the GNSS-PWV does not have a good linear relationship with AIRS-PWV. The slope (~ 0.64) of linear fitted trend and the correlation coefficient (R ~ 0.56) show a weak correlation between the two observations in the Pacific Ocean. From the bias value between the two observations, we can see that the GNSS-PWV is relatively slightly overestimated compared to the AIRS-PWV. This indicates that AIRS-PWV has a negative bias. Qin et al. [40] suggested that a negative bias of the AIRS-PWV is highly correlated to the temperature bias. The RMS error of AIRS-PWV for GNSS-PWV is larger than that of radiosonde-PWV for GNSS-PWV. The reason why the accuracy of AIRS-PWV is relatively low might be related to the spatial resolution of satellite data. The AIRS has a spatial footprint of 13.5 km at nadir from the about 705 km orbit [31]. Some studies reported the accuracy of the AIRS-PWV [41,42]. Recently, Heng and Jiang [43] reported that a correlation coefficient between AIRS-PWV and radiosonde-PWV is about 0.55 ~ 0.58 in the Lee of the Tibetan Plateau with a 95% confidence level. As a result, GNSS-PWV has higher accuracy than AIRS-PWV if radiosonde-PWV is used as a reference. 

## 5. Summary and Conclusions

In this study, we estimated PWV from shipborne GNSS observations collected by the low-cost GNSS receiver at the vessel ISABU in the Pacific Ocean. The GNSS derived-PWV was compared to radiosonde-PWV and AIRS-PWV. The PWV retrievals from the ground GNSS stations were in good agreement with radiosonde observations. When the GNSS-PWV was compared to AIRS-PWV, the biases and RMS errors were relatively large. Moreover, the satellite’s AIRS-PWV tended to be overestimated on the ground.

The research vessel ISABU operated in the Pacific Ocean for scientific research during the period from DOY 242 to DOY 264, 2018. We used the shipborne GNSS data for this period to calculate the PWV on the vessel’s paths. The estimated shipborne GNSS-PWV was also compared to radiosonde-PWV and AIRS-PWV. The bias and RMS error between shipborne GNSS-PWV and radiosonde-PWV were −1.48 and 5.22 mm, respectively. These errors on the sea surface are slightly larger than those on the ground. This may be related to multipath errors. The RMS value of MP1 on the sea surface was significantly larger than that of MP1 on the ground. However, the shipborne GNSS-PWVs are still coincident with radiosonde observations. In addition, we found that there was a strong correlation (R~0.85) between the two different observations. On the other hand, shipborne GNSS-PWV did not have a clear linear relationship with AIRS-PWV. The correlation coefficient between the two observations is about 0.56. It indicates that shipborne GNSS-PWV has a weak correlation with AIRS-PWV. The remarkable thing is that AIRS-PWV was slightly underestimated compared to the shipborne GNSS-PWV. In previous ground tests, AIRS-PWV was overestimated compared to GNSS-PWV.

The shipborne GNSS-PWV showed relatively weak agreement with other observations. This may be because carrier phase multipath error on the sea surface is significantly larger than that on the ground. The performance of GNSS equipment can also have a significant impact on PWV accuracy. Even using the low-cost GNSS equipment on the ocean, GNSS-PWV showed good agreement with other observation. However, the rapid movement of the vessel during radiosonde observations can cause a significant error between different observations due to differences in observation regions and the rolling of the vessel.

From the results presented in this paper, we suggest that the low-cost shipborne GNSS has sufficient potential to derive PWV with a moving vessel using the kinematic PPP solution. In particular, there is not much information about precise PWV in the ocean. Therefore, precise GNSS observations, along with the vessel’s route, are very significant to analyze the distribution and changes of PWV in the troposphere.

## Figures and Tables

**Figure 1 sensors-20-04261-f001:**
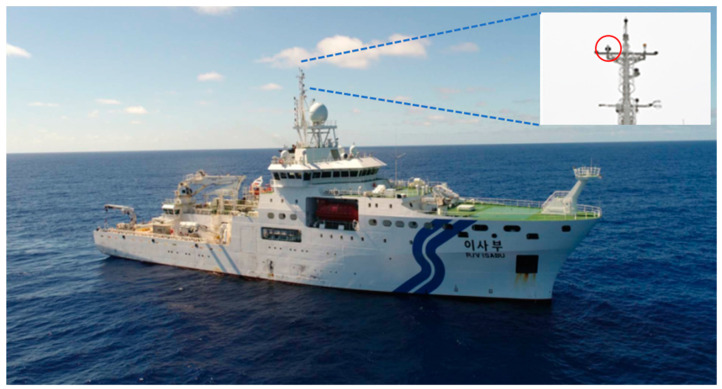
The Global Navigation Satellite System (GNSS) antenna (red circle) installed on the mast of the research vessel ISABU.

**Figure 2 sensors-20-04261-f002:**
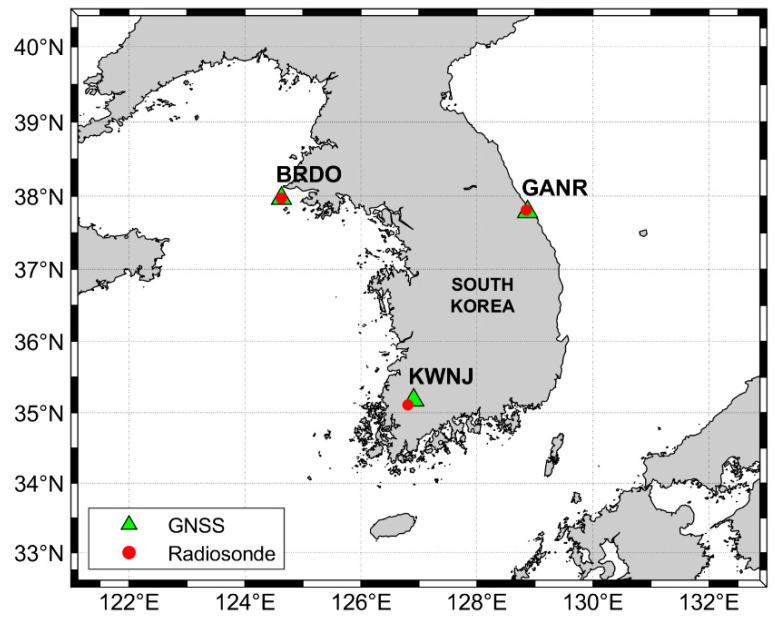
Location of GNSS stations and radiosonde. The green triangles and red dots in South Korea represent three GNSS stations (Backryeong-do (BRDO), Gangrenug (GANR), and Kwangju (KWNJ)) and adjacent radiosonde stations, respectively.

**Figure 3 sensors-20-04261-f003:**
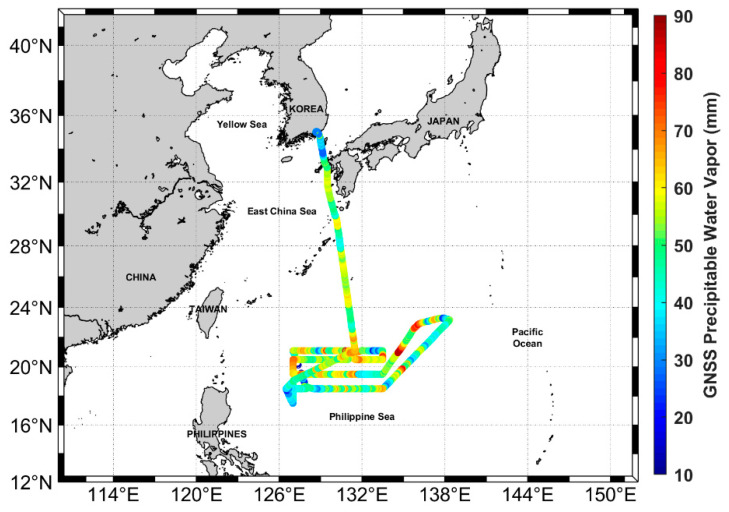
The PWV estimated from the shipborne GNSS observations with the vessel ISABU trajectory. The original purpose of this cruise campaign was to observe warm pool air–sea heat interaction supported to develop Typhoons. While a typhoon approaches the observation area, the vessel stayed in a safe zone of the neighboring countries.

**Figure 4 sensors-20-04261-f004:**
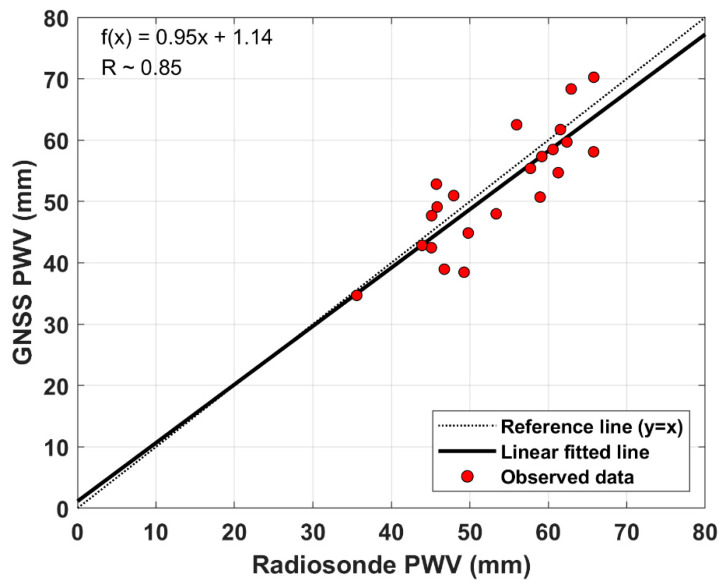
Comparison of shipborne GNSS-PWV against radiosonde observations. The slope of linear fitted trend the correlation coefficient (R) between GNSS-PWV and radiosonde-PWV is calculated with a linear regression model.

**Figure 5 sensors-20-04261-f005:**
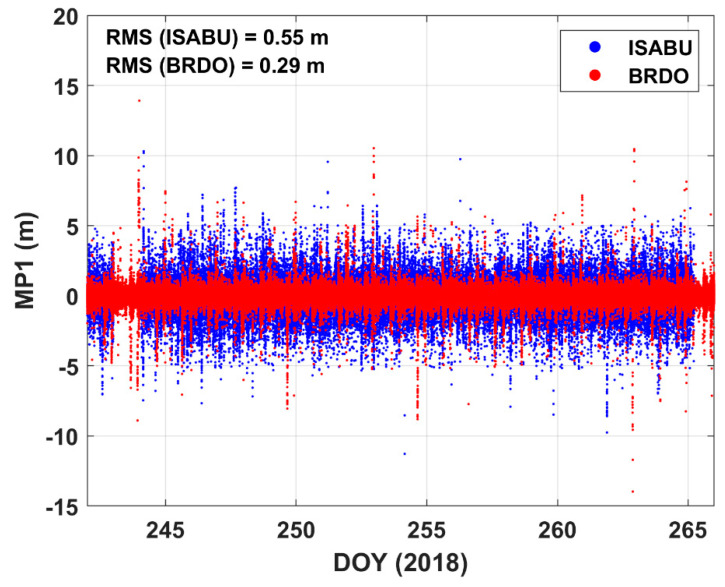
Time series of GPS L1 code multipath (MP1) errors. The blue dots and red dots represent MP1 at the vessel ISABU and MP1 at the ground BRDO station, respectively.

**Figure 6 sensors-20-04261-f006:**
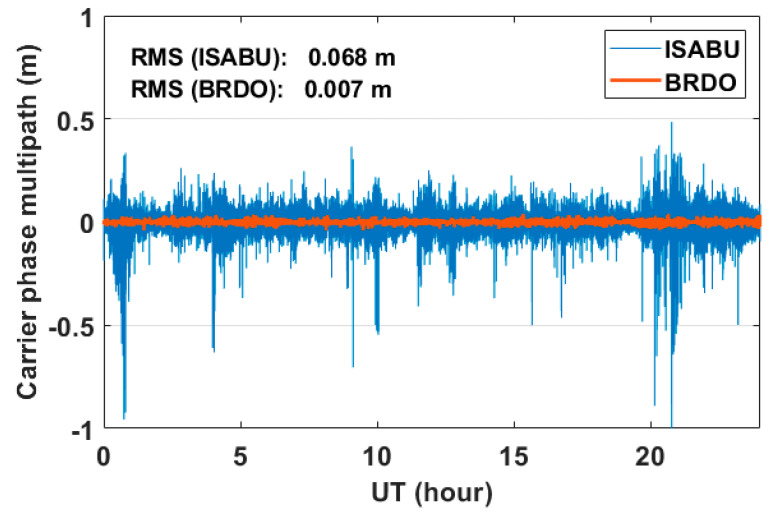
Time series of GPS L3 carrier phase multipath errors on DOY 242, 2018. The solid blue line and solid brown line represent carrier phase multipath errors at the vessel ISABU and at the ground BRDO station, respectively.

**Figure 7 sensors-20-04261-f007:**
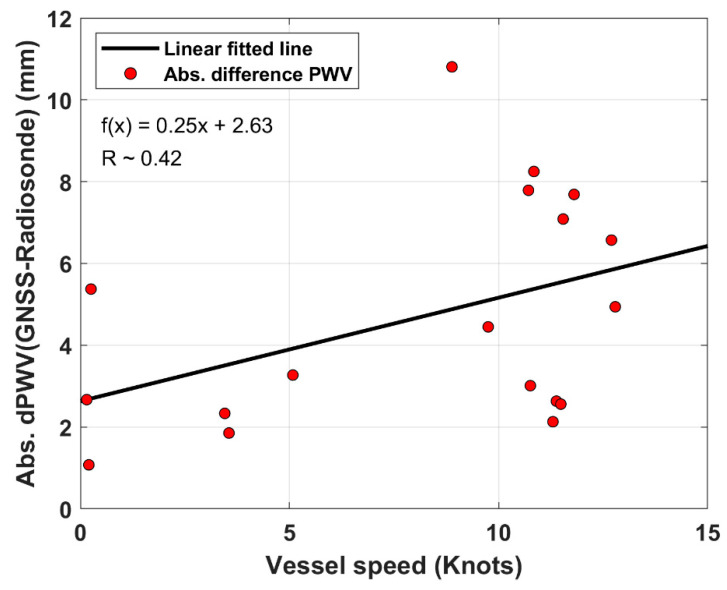
Correlation between absolute PWV difference (GNSS – Radiosonde) and vessel ISABU speed. The red dots denote the absolute difference between GNSS-PWV and radiosonde-PWV. The black solid line represents the relationship between the two variables by fitting a linear regression approach. The relation between the two variables tends to increase with the vessel’s speed increase.

**Figure 8 sensors-20-04261-f008:**
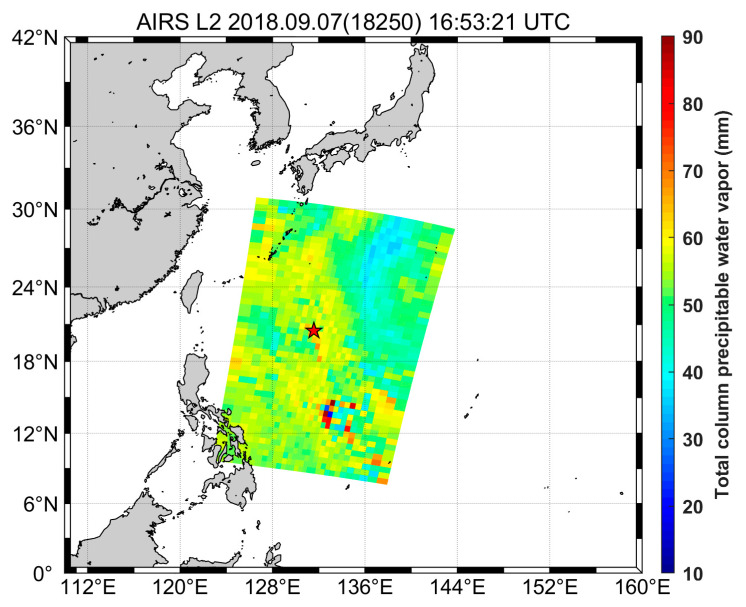
AIRS-PWV observation at the specific time and the location of the vessel ISABU (red star) at that time. The AIRS-PWV was averaged by extracting the observed values within the latitude ± 0.5° and longitude ± 0.5° from the ship’s position.

**Figure 9 sensors-20-04261-f009:**
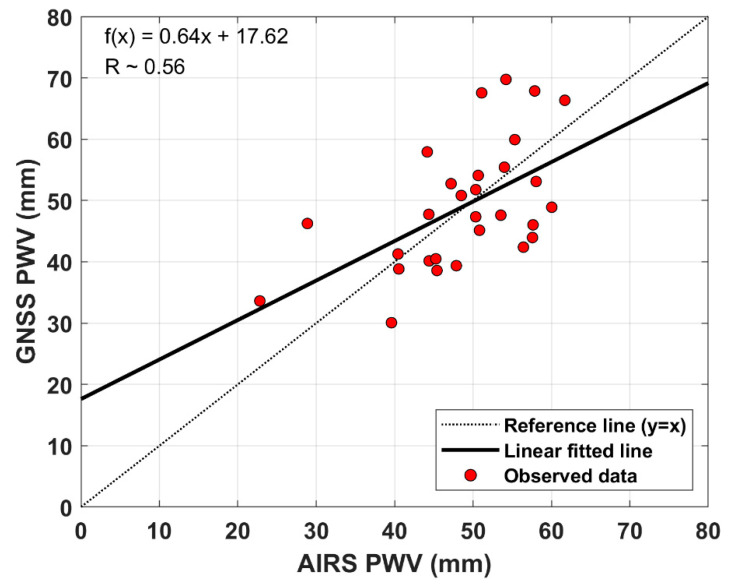
Comparison of shipborne GNSS-PWV against AIRS-PWV. The slope of linear fitted trend the correlation coefficient between GNSS-PWV and AIRS-PWV is calculated with a linear regression model.

**Table 1 sensors-20-04261-t001:** Models and methods for the GNSS kinematic precise point positioning (PPP).

Item	Models/Methods
Observations	Un-differenced Ionospheric Free Linear Combination (IFLC)
Signal	GPS: L1/L2
Elevation cutoff	10°
Weight for observations	Elevation-dependent angle stochastic model ($\sigma^{2}=1/\sin\left(E\right))$, E is the elevation angle of the satellite
Sampling rate	30 sec
Satellite orbit and clock, Earth rotation parameters	IGS final products
Satellite PCO/PCV	IGS14.atx
Receiver PCO/PCV	IGS14.atx
Phase wind-up	Wu et al. [29]
Solid earth tide, ocean tide, pole tide	IERS conventions 2010
Receiver clock	Estimated by Gauss-Markov model
Ionosphere	Ionospheric 1st-order delay is eliminated by IFLC
Troposphere	Estimated with tropospheric ZWD, $G_{N}$, and $G_{E}$
Tropospheric mapping function	GMF
Ambiguity	Float solutions

**Table 2 sensors-20-04261-t002:** The GNSS receiver and antenna types at the three selected stations on the ground.

Site Name	Receiver Type	Antenna Type	Latitude (Deg.)	Longitude (Deg.)
BRDO	Trimble NetR9	TRM55971.00	37.96 N	124.62 E
GANR	JAVAD TRE_G3T	JAVRINGANT_DM	37.78 N	128.87 E
KWNJ	Trimble NetR9	TRM59800.80	35.17 N	126.91 E

**Table 3 sensors-20-04261-t003:** Statistical comparison of precipitable water vapor (PWV) retrieval from ground-based three GNSS stations, radiosonde retrieved-PWV, and Atmospheric Infrared Sounder (AIRS) measurements.

GNSS Stations	Statistics		Comparison	
		GNSS–Radiosonde	Distance (km)	GNSS–AIRS
BRDO	Bias (mm)	1.27	0.01	−2.64
	RMS (mm)	2.47		4.38
GANR	Bias (mm)	−0.60	2.60	−5.19
	RMS (mm)	2.73		6.24
KWNJ	Bias (mm)	−4.13	10.60	−7.25
	RMS (mm)	4.40		7.40

**Table 4 sensors-20-04261-t004:** Comparisons between GNSS-PWV and radiosonde-PWV for a period from day-of-year (DOY) 242 to DOY 264, 2018 in the Pacific Ocean.

Observations (Numbers)	GNSS-PWV (mm)	Radiosonde-PWV (mm)	Difference (mm)
1	42.47	45.10	−2.63
2	34.70	35.59	−0.89
3	38.96	46.75	−7.79
4	55.40	57.73	−2.33
5	57.33	59.18	−1.85
6	42.84	43.91	−1.07
7	70.25	65.80	4.45
8	58.47	60.60	−2.13
9	59.70	62.37	−2.67
10	61.72	61.55	0.17
11	54.70	61.27	−6.57
12	68.34	62.92	5.42
13	62.51	55.97	6.54
14	58.09	65.78	−7.69
15	47.99	53.36	−5.37
16	38.46	49.27	−10.81
17	49.10	45.83	3.27
18	50.96	47.95	3.01
19	52.83	45.74	7.09
20	47.69	45.13	2.56
21	44.86	49.80	−4.94
22	50.71	58.96	−8.25

**Table 5 sensors-20-04261-t005:** Comparisons between GNSS-PWV and AIRS-PWV for a period from DOY 242 to DOY 264, 2018 in the Pacific Ocean.

Observations (numbers)	GNSS-PWV (mm)	AIRS-PWV (mm)	Difference (mm)
1	41.27	40.43	0.84
2	46.25	28.88	17.37
3	30.07	39.59	−9.52
4	47.34	50.34	−3.00
5	66.34	61.71	4.63
6	67.56	51.10	16.46
7	57.93	44.16	13.77
8	67.87	57.87	10.00
9	55.43	54.00	1.43
10	53.10	58.05	−4.95
11	59.93	55.33	4.60
12	43.96	57.58	−13.62
13	39.37	47.87	−8.50
14	51.77	50.35	1.42
15	38.58	45.40	−6.82
16	50.81	48.50	2.31
17	45.14	50.82	−5.68
18	40.15	44.40	−4.25
19	52.74	47.20	5.54
20	38.83	40.53	−1.70
21	40.49	45.25	−4.76
22	47.60	53.55	−5.95
23	46.03	57.65	−11.62
24	69.75	54.20	15.55
25	42.39	56.43	−14.04
26	54.09	50.66	3.43
27	48.90	60.04	−11.14
28	47.74	44.36	3.38
29	33.62	22.81	10.81

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
