# Peer review of "Precipitable Water Vapor Retrieval from Shipborne GNSS Observations on the Korean Research Vessel ISABU"

_sensors, 2020, doi:10.3390/s20154261_

Round 1

Reviewer 1 Report

This manuscript estimates precipitable water vapor from the shipborne GNSS observations.

The issues of the manuscript are shown as follows:

  • The novel contribution is not clear. This manuscript mainly introduces the satellite data, but the novel algorithm is not introduced very clear.
  • Some figures do not contain legend.

Reviewer 2 Report

The paper is clear and generally well written and show interesting results of PWV estimation by GNSS on a ship during an experimental campaign.

General remarks and questions:

1. Please specify the used PPP processing software, and its origin.

2. How are the radiosonde data completed for high altitude ?

3. Maybe use the same unit for PWV throughout the paper (mm or kg/m²).

4. A word about the meteorologic conditions during the expedition could be useful.

5. A typical sample image of AIRS data could provide us some insight about the spatial variability of the PWV on the area during the expedition.  

6. You are right to use multipath errors on the code for comparison purpose between earth and sea measurements, but it is not representative of the multipath error on the final result of PPP,  which is due mostly to multipath errors on the phase.

More specific remarks:

Line 13: "GNSS receivers are equipped with a research vessel" seems strange to me, I would have better :"GNSS receivers that the vessel is equipped with"

Line 22: "have a good linear relationship"-> "are not well linearly correlated" ?

Line 24: "it is revealed " -> "we show"

Line 37: the gnss has no high spatial resolution per se.

Line 38: "using the ground-based"

Line 41: you could add Raman LIDAR WV profiling.

Line 87-103: a simple reference presenting PPP could be enough. You don't use the equations in the paper.You should however mention which software you use.

Line 193: I can't quite understand " It can be sufficiently applied "

Line 219: 0.85 may be a "strong" correlation for two random set of data, but I would not use this word to qualify two data sets that are measurements of the same quantity.

Line 298: I would delete the two sentences, that only state the obvious.

Reviewer 3 Report

The paper describes the comparisons among precipitable water vapor (PWV) retrieved from shipborne GNSS observations, radiosonde observations and the atmospheric infrared sounder (AIRS) on NASA's aqua satellite, and discusses the possibilities of shipborne GNSS to obtain PWV data, which is very important to analyze the PWV distribution and variations globally in the troposphere. Howerver, I think some minor modifications are still needed to be improved before the paper is accepted  for publications. Some suggestions are as follows,

1, In Section 2.1, it is not clear how to obtain PWV from equation (1-4) and table 1 as suggested in the paper. As PWV data from radiosonde or AIRS retrieval is the integral water vapor vertically, GNSS data comes from different satellites in different zenith angle or incident angle. It is better to give more detail how to retrieve PWV from GNSS data (which parameters as shown in equation 1-4). In the table 1, why the elevation cutoff is 10 degree?

2, Two different unit of PWV are used in the paper, kg/m^2 or  mm, it is better to use the same unit in the paper, or give the value of kg/m^2 in mm if possible.

3, Some English wording problems, such as the mixture use of 'compare to' and 'compare with'(in the abstract), 'between A and B and C' (page 2 line 50) ,'this may be because ...' (page 11 line 303-304). It is better to ask native speaker to check the paper throughly.

4, Some explanations are not clear. Page 10 line 299, 'This may be due to a significant difference between the shipborne GNSS-PWV and AIRS-PWV', what's the big difference? temporal or spatial difference? Is is possible to average the shipline GNSS-PWV around 13.5km same as the footprint of AIRS-PWV before comparison as shown in Fig.6?

Page 10 line 275, what about the accuracy of the AIRS-PWV?

Page 11 line 306, what the mean of 'the rapid movement  or the rolling of the vessel'? Is there any such data to provide for reference to describe the rapid movement of the ship?

Round 2

Reviewer 1 Report

For the EKF PPP estimation, only the state vector  is introduced. However, the state model and measurement model are not introduced. This is very important for the PPP estimation.
